

# Creatinine is a biochemical marker for assessing how untrained people adapt to fitness training loads

Andrii Chernozub[1], Vladimir Potop[2], Georgiy Korobeynikov[3], Olivia Carmen Timnea[4], Oleg Dubachinskiy[1], Oksana Ikkert[5], Yuriy Briskin[5], Yuriy Boretsky[5] and Lesia Korobeynikova[3]

[1] Petro Mohyla Black Sea State University, Mykolaiv, Ukraine
[2] Faculty of Physical Education and Sport, Ecological University of Bucharest, Bucharest, Romania
[3] National University of Physical Education and Sport, Kyiv, Ukraine
[4] Romanian-American University, Bucharest, Romania
[5] Ivan Bobersky Lviv State University of Physical Culture, Lviv, Ukraine

Corresponding author
Vladimir Potop,
vladimir_potop@yahoo.com

## ABSTRACT

**Background**. To study the peculiarities of changes in creatinine concentration in blood serum of untrained men during the prolonged usage of training loads different in volume and intensity, and to determine the value of this biochemical marker for the assessment of adaptive body changes during fitness training.

**Methods**. We examined 50 untrained men aged 18–20 years who had no contraindications for practicing fitness. Taking into account the aim of the research, we divided these people into two groups: group A and group B. The research participants used training load regimes different in volume and intensity: representatives of group A used low intensity training load regime ($Ra = 0.53$) and representatives of group B used high intensity training load regime ($Ra = 0.72$). To assess the adaptive body changes in the examined contingent, we used the anthropometry method (circumference body size) and bioimpedansometry (body composition indicators). We also determined the features of adaptation-compensatory body reactions to different training loads by means of biochemical control of creatinine concentration in blood serum.

**Results**. The obtained results showed a significant increase in basal level of creatinine concentration in blood serum (by 17.6%) of group B representatives fixed after 3 months of practicing fitness in high intensity training load regime ($R_a = 0.72$). This group representative also revealed the most pronounced manifestation of adaptive body changes confirmed by the results of the maximal muscle strength growth ($1RM$) and circumference body size, which was almost 2.5 times higher than the results of group A representatives for the same period of time. The parameters indicating the load volume in a set ($Wn$) were almost 62.0 % higher in group A representatives in comparison with group B during all stages of control. Nonetheless, the highest creatinine concentration in blood serum (by 11.1 % ($p < 0.05$) was fixed in group A representatives in response to training load after 3 months of practicing. This fact testifies to the important role of the creatine phosphokinase mechanism of energy supply of muscular activity in the conditions of high volume and low intensity training load regime ($R_a = 0.53$).

**Conclusion**. The analysis of the results obtained during a series of experimental studies indicates the need and feasibility of using the indicator of basal level of creatinine concentration in blood during fitness training, especially in the conditions of high

intensity and low volume training load regime ($R_a = 0.72$), as an informative marker for assessing the process of long-term adaptation.

# INTRODUCTION

To ensure the effectiveness of biomedical control at all stages of training activity, the leading specialists in physical education and biology (*Chernozub, 2015*; *Cadegiani & Kater, 2019*; *Chlíbková et al., 2019*) are in constant search for the most informative markers of assessing adaptation-compensatory responses to physical stress stimuli. Due to the constant increase in training loads, the body undergoes significant metabolic changes to ensure the necessary level of training. However, long-term functioning of the body in conditions of constant muscular tension without appropriate mechanisms of training load regulation will lead not only to exhaustion of functional body reserves, but also to failure of adaptation (*Butova & Masalov, 2011*; *Kılıç et al., 2019*; *Kistner et al., 2019*).

The use of blood biochemical parameters for the diagnosis of athletes' functional state in the course of training activity allows determining the limits of adequacy of training load parameters to the primary level of body adaptation to stress stimuli of the appropriate character at the beginning of studies (*Fomin, Gorokhov & Tymoshenko, 2006*; *Rubio-Arias et al., 2019*). Taking into account the fact that training load in fitness has mainly anaerobic character, the study of creatine phosphokinase mechanism of energy supply of muscle activity, and peculiarities of its biochemical marker changes in blood serum, is a very topical issue (*Spada et al., 2018*; *Omassoli et al., 2019*).

Creatinine is one of the biochemical indicators of blood, which reflects the peculiarities of the creatine phosphokinase mechanism of energy supply of physical activity. Creatinine is known to be the end product in the chain of metabolic breakdown of creatine phosphate (*Shin et al., 2016*; *Zakalskiy, Stasyuk & Gonchar, 2019*). A number of researchers (*Fomin, Gorokhov & Tymoshenko, 2006*; *Butova & Masalov, 2011*; *Futorniy et al., 2016*) state that creatinine concentration in blood serum of a healthy person does not change, and its level depends on the muscle mass volume.

In the course of experimental studies, several specialists (*Butova & Masalov, 2011*; *Omassoli et al., 2019*) proved that the creatine phosphokinase mechanism of energy supply of muscle activity is really important for athletes specializing in high-intensity sports allowing considering the creatinine concentration index in blood serum to be a biochemical marker of the term adaptation. Accordingly, the control of creatinine concentration dynamics in blood serum during training load, in some cases, indicates the state of the whole creatine phosphokinase system, its effect on increasing the body functional reserves, the level of resistance to stress stimuli, and the economy of energy (*Papassotiriou & Nifli, 2018*; *Spada et al., 2018*; *Stajer, Vranes & Ostojic, 2018*).

*Castro et al. (2019)* and *Tang, Chan & Kuo (2014)* studied the reaction of creatinine to a physical stress stimuli. *Wang et al. (2016)* and *Spillane et al. (2009)* researched the impact of using substances containing creatinine on adaptive body changes during physical training. However, we found no studies concerning the use of creatinine as a biochemical marker and an indicator for assessing adaptation-compensatory body reactions in people practicing fitness.

The purpose of the research was to study the peculiarities of changes in creatinine concentration in blood serum of untrained men during the prolonged usage of training loads different in volume and intensity; and to determine the value of this biochemical marker for the assessment of adaptive body changes during fitness training.

## MATERIALS & METHODS

### Participants

We examined 50 untrained men aged 18–20 years who had no contraindications for practicing fitness. Taking into account the aim of the research, we divided these people into two groups: group A and B. The research participants used training load regimes different in volume and intensity: representatives of group A used low intensity training load regime ($R_a = 0.53$) and representatives of group B used high intensity training load regime ($R_a = 0.72$).

For the accuracy of the studies, participants were selected in such a way that the initial parameters of their anthropometric body indicators and power capabilities of the organism did not have significant differences between the representatives of the surveyed groups.

The experimental study was approved by Petro Mohyla Black Sea National University Ethics Committee for Biomedical Research in accordance with the Ethical Standards of the Helsinki Declaration (ecbr5-10-2019). The participants gave written consent to the study in accordance with the recommendations of the Biomedical Research Ethics Committees (*WHO, 2000*).

An equipped and certified medical laboratory was used for medical examination and the comprehensive biochemical laboratory control (16 indicators) of untrained people who participated in the research.

### Methods

Assessment of maximal strength of muscular groups (1 RM, kg) of participants during 3 months of fitness training was carried out during the performance of control exercises: bench press on the block of the training equipment "Smith machine", sitting dumbbellbench press, moving hands with dumbbells, extension of arms on the block standing, lifting dumbbells on biceps standing. The assessments were made before the training session using the standard method and technique of performing exercises (*Hatfield, 1993*). After the warm-up session, participants were given 3 attempts to determine the maximum strength development parameters for a particular muscle group. We put down the result of the best attempt to the study protocol.

To determine the characteristics of training load regimes, we used the integral method of quantitative estimation of load capacity in power fitness (*Chernozub et al., 2018*). The

training load index ($R_a$), which reflects the features of the training load regimes in fitness, was calculated according to the formula:

$$R_a = R_{\max} - (nQtf_o),$$

where $R$max is the maximum training load index with the value $R_{\max} = 1$;

n is a certain number of repetitions in a separate working set;

Q is the conditional amplitude coefficient. Depending on the orientation of the training process, physical exercises are performed with full or partial amplitude. The quantitative values of this indicator are in the range: $0.8 \leq Q \leq 1$.

t is the duration of one repetition (s). This figure is within $3 \leq t \leq 9$.

$f_o$ is the empirical coefficient obtained by multiple regression analysis, where the values of the independent variables $(n, t, Q, m)$ are found experimentally, the value of $f_o = 0.0098$ 1/s.

The projectile working mass suitable for an athlete in the given conditions of the training load regime was calculated according to the formula:

$$m = R_a 1RM,$$

where m is the projectile working mass (kg), which can be lifted by a person in each repetition. The number of repetitions depends on the peculiarities of the training load regime;

$R_a$ is the training load index;

1RM is the maximum projectile mass (kg), which a person can overcome while performing control exercises but only once.

The volume of training load in fitness was calculated according to the formula:

$$Wn = mN_{\max},$$

where Wn is the amount of training load in a set (kg) which is the total projectile working mass that a person lifted while using a certain training load regime with the maximum number of repetitions in a set to full muscle fatigue;

m is the projectile working mass (kg) which a person can lift in each repetition;

$N_{\max}$ is the maximum number of repetitions in a working set which a person can perform in a certain training load regime.

To determine the initial parameters of body composition indices and to study the characteristics of their dynamics in the process of prolonged fitness training, we used the method of bioimpedansometry. This non-invasive, biophysical method is based on measuring the electrical resistance of biological tissues of the body. The body composition is determined in the process of computer processing of the obtained results. Using this method allows calculating the following indices: body fat percentage (BF, %); fat-free mass (FFM, kg); body cell mass (BCM, kg); body mass index (BMI, c.u.). To evaluate the above indicators, we used a bioimpedance analyzer: diagnostic computerized hardware-software complex KM-AP-01 developed by "Diamond—AST" (body composition analyzer) (VYUSK. 941118.001 RE) (*Martyrosov, Nikolayev & Rudnev, 2006*).

The circumference body size of the shoulder, chest, and thighs were measured in participants of the research with the help of a centimeter tape.

The creatinine concentration in the blood serum was checked by means of the kinetic method on the equipment of "High Technology Inc" (USA) with a set of reagents PRESTIGE 24i. The blood sampling procedure was performed according to the general requirements of biomedical research (*Tietz, 1995*).

Blood samples were numbered, described in the appropriate way and delivered to the clinical laboratory. The physiologically acceptable creatinine concentration in blood serum of healthy men is in the range from 70 to 110 µmol/l. Laboratory studies were performed at the beginning of the experiment and after 3 months of fitness training.

## Experimental design

The research was conducted in several stages:

- in the first stage we developed a training load regime ($R_a = 0.72$) for group B with corresponding changes in the duration of muscle activity time periods. We reduced the rest period between sets by 33% and increased the time for performing movement phases by 50%, which resulted in reducing the number of repetitions in a working set almost twice and increasing the projectile working mass ($m$) by 14.3%. It also caused the fatigue acceleration in working muscle groups due to repeated physical loads and the lack of complete restoration of energy supply systems. The training load regime for group A ($R_a = 0.53$) included training load parameters considered to be effective and contribute to the accelerated growth of muscle mass by experts in fitness, bodybuilding, powerlifting (*Hatfield, 1993*; *Butova & Masalov, 2011*; *Titova et al., 2018*). Both training load regimes are presented in Table 1.

- in the second stage we studied changes in parameters of maximum muscular strength, circumference body size within 3 months of using different in volume and intensity load regimes in order to determine the most optimal conditions of muscular activity to accelerate the processes of adaptation to physical stress stimuli; at the same time we investigated the peculiarities of changes in creatinine concentration in blood serum, both at rest and after training, to assess the creatine phosphokinase mechanism of energy supply of muscular activity while using different load regimes of fitness training.

There were 3 training sessions per week in the course of research. The duration of one training session was no more than 40 min, regardless of the peculiarities of training load regimes. No more than 3–4 muscle groups were loaded in each training. 2–3 training exercises with 4 sets in each exercise were applied to each muscle group. Training exercises were performed with the barbell, dumbbells on the training equipment following the defined techniques.

- in the third stage we studied the changes in parameters of maximum muscular strength, circumference body size within 3 months of using different in volume and intensity training load regimes. We also studied the most optimal conditions of muscular activity to accelerate the processes of adaptation to physical stress stimuli. We investigated the peculiarities of changes in creatinine concentration in blood serum of both groups' representatives to

**Table 1** Structural features of training load regimes used by representatives of study groups during fitness training.

| Training load index | Training load regimes | |
| --- | --- | --- |
| | Training load regime for group A | Training load regime for group B |
| Amplitude of performing exercise, % | Full (100%) with projectile working mass fixation in a peak point | Partial (90% from maximum) |
| Conditional amplitude coefficient ($Q$) | 1 | 0.8 |
| Duration of movement phases | Concentric phase—2 s; Eccentric phase—3 s; Projectile working mass fixation in a peak point—1 s | Concentric phase—3s; Eccentric phase—6s; |
| Duration of one repetition ($t$, s) | 6 | 9 |
| Duration of rest between sets, s | 60–70 | 40–45 |
| Number of repetitions in one working set | 8–10 | 4–5 |
| Maximalldurationofworkinone set (s) | 48–60 | 36–45 |
| Projectile working mass (m), % from maximum (1RM) | 63–68 | 72–74 |
| Training load index ($R_a$) | 0.53 | 0.72 |
| Level of intensity and volume of training load | Low intensity and high volume | High intensity and low volume |
| Duration of a training, min | 30–35 | 30–35 |

assess the creatine phosphokinase mechanism of energy supply of muscular activity in different training load regimes;

- in the fourth stage we compared the results of the dynamics of the circumference body size, reflecting the muscle mass growth rate in both groups representatives, with changes in the basal level of creatinine concentration in blood serum and the changes in its concentration in response to stress stimuli. The obtained results will allow determining the level of body resistance to the proposed training load, and the effectiveness of using creatinine as an informative marker of assessing long-term adaptation during fitness training.

## Statistical analysis

Statistical analysis of the study results was performed using the IBM * SPSS * Statistics 22 software package (*StatSoftInc.*, USA). Descriptive statistics methods were used to calculate the arithmetic mean and the error of the mean. Non-parametric Wilcoxon criterion was used to assess the reliability of pairwise differences; Friedman's ANOVA was used to analyze repeated measurements (e.g., *Nasledov, 2013*).

## RESULTS

Table 2 presents the results and characteristics of the dynamics of mean group results of training load indices and maximum power capabilities of both groups' participants during 3 months of studies.

We conducted an analysis of variance to confirm the peculiarities of the influence of developed training load regimes on the functional state of the studied groups' participants. According to the Leuven test, the variances of the analyzed indicators were excellent. In

**Table 2  Mean group results of training load indices fixed during 3 months of fitness training, $n = 50$.**

| Training load index | Group | Period of observation, months | | | $\chi^2, p$ $df = 2$ |
|---|---|---|---|---|---|
| | | Initial data | 1.5 | 3 | |
| Maximum projectile mass (1RM), kg | A | $9.49 \pm 0.21$ | $10.55 \pm 0.15^*$ $Z = -3.9;$ $p < 0.000$ | $10.81 \pm 0.18^*$ $Z = -2.4;$ $p < 0.014$ | $\chi^2 = 40.9$ $p < 0.000$ |
| | B | $11.05 \pm 0.32$ | $13.19 \pm 0.39^*$ $Z = -4.5;$ $p < 0.000$ | $15.03 \pm 0.33^*$ $Z = -3.9;$ $p < 0.000$ | $\chi^2 = 47.5$ $p < 0.000$ |
| Projectile working mass (m), kg | A | $15.67 \pm 0.47$ | $17.63 \pm 0.56^*$ $Z = -4.1;$ $p < 0.000$ | $18.74 \pm 0.63^*$ $Z = -4.0;$ $p < 0.000$ | $\chi^2 = 46.9$ $p < 0.000$ |
| | B | $18.40 \pm 0.72$ | $22.28 \pm 0.83^*$ $Z = -4.6;$ $p < 0.000$ | $27.72 \pm 0.76^*$ $Z = -4.3;$ $p < 0.000$ | $\chi^2 = 50.0$ $p < 0.000$ |
| Amount of training load in a set ($Wn$), kg | A | $134.75 \pm 5.05$ | $155.28 \pm 6.66^*$ $Z = -4.1;$ $p < 0.000$ | $181.24 \pm 7.75^*$ $Z = -3.8;$ $p < 0.000$ | $\chi^2 = 46.5$ $p < 0.000$ |
| | B | $83.20 \pm 4.70$ | $101.20 \pm 5.59^*$ $Z = -4.4;$ $p < 0.000$ | $125.48 \pm 5.73^*$ $Z = -4.3;$ $p < 0.000$ | $\chi^2 = 49.6$ $p < 0.000$ |

**Notes.**
*Difference compared with previous results is significant according to Wilcoxon criterion ($p < 0.05$); $df$ is the number of degrees of freedom; $p$ is the level of significance.

addition, the indicators were not distributed according to normal law, which did not allow using the parametric ANOVA of repeated measures. That is why we used a Friedman repeated measures ANOVA. It is a nonparametric analogue of ANOVA for conducting analysis of repeated measurements related to the same person. To clarify the obtained results, pairwise comparisons were made using Wilcoxon's criterion in different periods of observation (Tables 2–3).

The analysis of the obtained results showed that in conditions of long-term usage of training load regime for group B, 1 RM (the index of maximal muscular strength) grew almost 3 times faster than the indices fixed in group A.

The parameters of the projectile working mass used by men in group B increased during 3 months of fitness training by 50.6% ($p < 0.05$) compared to baseline data. The studied indices of participants in group A increased by 19.6% ($p < 0.05$) for the same period of time.

Having analyzed the control results of the indicator of amount of training load in a set ($Wn$) we noticed a certain pattern of changes, depending on the characteristics of training load regimes (Table 2). Thus, despite the higher baseline level of power capabilities development (by 16.4% compared to opponents), participants of group B had lower initial parameters of $Wn$ by 38.2% ($p < 0.05$) than group A representatives.

We found out that despite the significant (about 51.0%) increase in the load volume parameters in a set, the results of the main group representatives were lower by 6.8% at the end of the study than the baseline data found in the control group due to the increase of the body power capabilities by 36.0%.

**Table 3 Parameters of morphometric body indicators of the participants of the surveyed groups during 3 months of fitness training, $n = 50$.**

| Parameters | Group | Period of observation, months | | | $\chi^2, p$ $df = 2$ |
|---|---|---|---|---|---|
| | | Initial data | 1.5 | 3 | |
| Shoulder circumference size, cm | A | $30.92 \pm 0.47$ | $32.26 \pm 0.52^*$ $Z = -4.4;$ $p < 0.000$ | $32.76 \pm 0.52^*$ $Z = -5.0;$ $p < 0.000$ | $\chi^2 = 50.0$ $p < 0.000$ |
| | B | $29.88 \pm 0.39$ | $32.12 \pm 0.40^*$ $Z = -4.6;$ $p < 0.000$ | $33.38 \pm 0.41^*$ $Z = -4.4;$ $p < 0.000$ | $\chi^2 = 50.0$ $p < 0.000$ |
| Chest circumference size, cm | A | $97.36 \pm 1.23$ | $98.12 \pm 1.19$ $Z = -4.5;$ $p < 0.000$ | $98.40 \pm 1.06$ $Z = -1.6;$ $p > 0.05$ | $\chi^2 = 23.0$ $p < 0.000$ |
| | B | $92.88 \pm 0.74$ | $95.36 \pm 0.76^*$ $Z = -4.5;$ $p < 0.000$ | $97.10 \pm 0.75^*$ $Z = -4.5;$ $p < 0.000$ | $\chi^2 = 50.0$ $p < 0.000$ |
| Thigh circumference size, cm | A | $51.58 \pm 0.63$ | $52.58 \pm 0.52^*$ $Z = -3.9;$ $p < 0.000$ | $52.58 \pm 0.50$ $Z = -0.0;$ $p > 0.05$ | $\chi^2 = 28.8$ $p < 0.000$ |
| | B | $48.60 \pm 1.18$ | $50.68 \pm 0.99^*$ $Z = -4.4;$ $p < 0.000$ | $52.36 \pm 0.94^*$ $Z = -3.8;$ $p < 0.000$ | $\chi^2 = 47.6$ $p < 0.000$ |
| BMI (body mass index), c. u. | A | $22.50 \pm 0.57$ | $22.21 \pm 0.51^*$ $Z = -2.7;$ $p < 0.006$ | $22.21 \pm 0.42$ $Z = -0.0;$ $p > 0.05$ | $\chi^2 = 0.5$ $p > 0.05$ |
| | B | $21.32 \pm 0.63$ | $21.40 \pm 0.64^*$ $Z = -2.4;$ $p < 0.014$ | $21.64 \pm 0.63^*$ $Z = -2.3;$ $p < 0.020$ | $\chi^2 = 22.6$ $p < 0.000$ |
| FFM (fat-free mass), kg | A | $55.78 \pm 1.22$ | $55.92 \pm 1.20$ $Z = -0.2;$ $p > 0.05$ | $56.47 \pm 1.10$ $Z = -1.3;$ $p > 0.05$ | $\chi^2 = 2.4$ $p > 0.05$ |
| | B | $55.56 \pm 0.81$ | $56.60 \pm 0.73$ $Z = -1.7;$ $p > 0.05$ | $57.52 \pm 0.84^*$ $Z = -2.8;$ $p < 0.004$ | $\chi^2 = 28.8$ $p < 0.000$ |
| BF (body fat), % | A | $19.26 \pm 0.80$ | $18.06 \pm 0.84^*$ $Z = -3.8;$ $p < 0.000$ | $17.37 \pm 0.76$ $Z = -1.3;$ $p > 0.05$ | $\chi^2 = 10.1$ $p < 0.006$ |
| | B | $15.89 \pm 0.99$ | $14.59 \pm 0.87$ $Z = -1.7;$ $p > 0.05$ | $14.27 \pm 0.82$ $Z = -1.3;$ $p > 0.05$ | $\chi^2 = 13.1$ $p < 0.005$ |
| BCM (body cell mass), kg | A | $36.50 \pm 0.87$ | $36.05 \pm 0.84$ $Z = -2.1;$ $p > 0.05$ | $37.03 \pm 0.73$ $Z = -1.8;$ $p > 0.05$ | $\chi^2 = 22.6$ $p < 0.000$ |
| | B | $37.04 \pm 0.45$ | $37.05 \pm 0.46$ $Z = 0.0;$ $p > 0.05$ | $37.49 \pm 0.48^*$ $Z = -2.8;$ $p < 0.004$ | $\chi^2 = 22.1$ $p < 0.000$ |

**Notes.**

*Difference compared with previous results is significant according to Wilcoxon criterion ($p < 0.05$); $df$ is the number of degrees of freedom; $p$ is the level of significance.

The results, recorded within 3 months of the study, showed that the indicator of amount of training load in a set increased by 51.0% ($p < 0.05$) in group B, due to the body strength capabilities increasing by 36.0%. At the same time, the mean group results of $Wn$ found in

the men of group B at the end of the study were 6.8% lower than the data fixed in group A representatives at the beginning of the experiment.

Thus, the obtained results indicate that high intensity and low volume training load regime for group B most effectively influence the increase in the maximum muscular strength of men of this age group. At the same time, the results of the research may allow us to find one of the effective ways to stop the process of constant increase of the volume of training loads (first of all, the amount of work). The solution of this problem will provide athletes with an optimal stress stimulus for the process of long-term adaptation and will help to reduce the level of injury.

The results of the features of circumference body size and body composition indicators of the examined contingent within 3 months of studies are presented in Table 3.

The results analysis showed that group A representatives who used low intensity and high-volume training load regime ($R_a = 0.53$) for 3 months, increased their shoulder circumference size by 5.9% ($p < 0.05$) compared to the initial data. The controlled indicator increased by 11.7% ($p < 0.05$) in the men of group B who used high intensity and low volume training load regime ($R_a = 0.72$) for the same period of time.

The chest circumference size showed a slight positive tendency to increase in the men of group A by 1.1% ($p > 0.05$) while the men of group B showed a significant increase of the studied indicator by 4.5% ($p < 0.05$) for the same period of time.

The results revealed during the control of the dynamics of the thigh circumference size indicated an effective increase by 7.7% ($p < 0.05$) in the men of group B during all stages of the study. However, there was 1.9% ($p < 0.05$) increase in thigh circumference size among participants of group A only during the first 1.5 months of the study, and then no further positive trends were observed.

The analysis of the body composition indices in the experimental studies showed that BMI increased by 1.5% ($p < 0.05$) during the study only in the men of group B. There was a decrease of the investigated index in group A representatives indicating significant loss of energy and manifestation of compensatory reactions to a stress stimulus.

The results revealed during all stages of the study indicated that fat-free body mass index increased by 4.4% ($p < 0.05$) in the men in group B compared with initial data. Group A results also reflected positive dynamics with almost 50% less progression.

Thus, based on the results of changes in morphometric body parameters, it can be argued that using training load regime B in fitness training contributes to the most pronounced adaptation body changes in untrained men aged 18–20 years.

We conducted several studies of the peculiarities of changes in creatinine concentration in blood serum in conditions of long-term using different training load regimes and presented the results of laboratory control in Fig. 1.

The results of analysis showed that at the beginning of the studies the level of creatinine concentration in blood serum of both group representatives showed a slight tendency to decrease in response to the load stimuli compared to the state of rest, regardless of the peculiarities of training load regimes. The obtained results indicated that, at this stage of the study, there was no activation of urgent adaptation to meet the growing needs of the organism for energy supply due to creatine phosphokinase mechanism, taking into

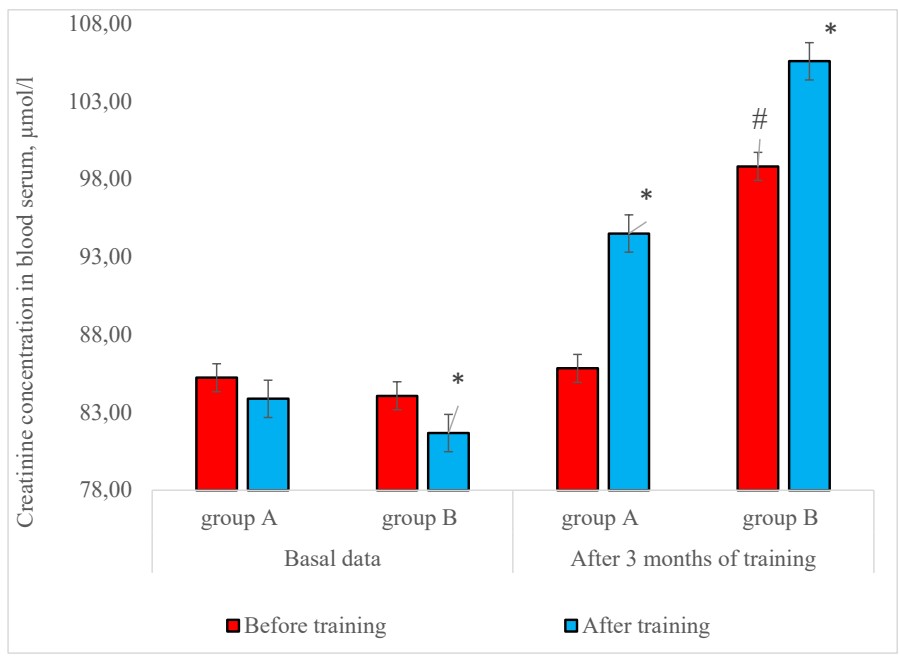

**Figure 1** **Changes in the creatinine concentration in blood serum of men in the study groups, $n = 50$.**
*—$p < 0.05$, compared to before training data; #—$p < 0.05$, compared to the previous results of basal creatinine level.

account the primary level of organism functionality of the examined contingent and the peculiarities of the proposed training load regimes (*Fomin, Gorokhov & Tymoshenko, 2006*; *Rubio-Arias et al., 2019*). Considering the level of training of the surveyed contingent (both group representatives were not engaged in fitness before the research), the initial level of maximal muscle strength development we found out that the parameters of intramuscular and intermuscular coordination were not high enough. In the given conditions of muscular activity, the energy supply was conducted mainly due to lactate anaerobic mechanism.

The results of biochemical control registered in group A after 3 months of fitness training showed an increase by 11.1% ($p < 0.05$) in creatinine concentration in blood serum after training compared to the state of rest. This fact testifies to the process of body adaptation to training loads of high volume and low intensity ($R_a = 0.53$) by essential usage of creatine phosphokinase mechanism of energy formation. Moreover, the basal level of the investigated blood index remained almost unchanged.

Analysis of the biochemical control results indicates that after 3 months of studies the mean group indicator of creatinine concentration in blood serum of group B participants increased by 6.8% ($p < 0.05$) compared to before training data. Comparative analysis of the biochemical control results of changes in creatinine concentration in blood serum of both groups showed an increase in the level of body resistance to power load in the men of group B. Accordingly, the most pronounced adaptation effect in the conditions of fitness training was observed when using developed by us high intensity and low volume training load regime ($R_a = 0.72$).

The results of basal dynamics of creatinine concentration in blood serum of group B participants indicated an increase of the studied index by 17.6% ($p < 0.05$) after 3 months of fitness training compared with the initial data. These changes showed that muscle growth increased the value of creatine phosphokinase energy supply, and the level of metabolic capacity of its reaction (*Spada et al., 2018*; *Stajer, Vranes & Ostojic, 2018*).

## DISCUSSION

One of the most debated problems of the optimization of biomedical control of the evaluation of effectiveness of adaptive body changes in people with different levels of training during fitness training, is the problem of determining the most informative biochemical markers. These markers will ensure establishing the load adequacy to functional body capabilities, and the result of training. Improvement of the system of fitness training occurs due to optimization of the system of control, and search for new mechanisms of training activity correction. These changes take place due to the development of effective and safe training load regimes, allowing quickly increase the level of body resistance to physical stress stimuli. These changes also contribute to the reduction of overtraining and adaptation failures (*Chernozub et al., 2018*; *Spada et al., 2018*; *Kistner et al., 2019*; *Omassoli et al., 2019*).

Studying the features of biochemical control in the process of monitoring the functional state during fitness training, a number of researchers (*Futorniy et al., 2016*; *Stajer, Vranes & Ostojic, 2018*; *Kılıç et al., 2019*) stated that the nature of changes in creatinine concentration clearly reflected the course of adaptation-compensatory responses to physical stimuli in conditions of creatine phosphokinase mechanism of energy supply of intense muscular activity.

The obtained results, which at the beginning of the survey showed a decrease in creatinine concentration in blood serum of untrained men of both groups, regardless of the peculiarities of the training load regimes, coincide with the data of experimental studies, used for sprint distance running (200 m, 400 m, 800 m) as training loads (e.g., *Butova & Masalov, 2011*). However, a decrease in creatinine concentration of physically healthy people was also observed in response to aerobic exercise associated with long distance running (over 54 km) (e.g., *Rubio-Arias et al., 2019*). At the same time, our findings contradict the results of studies conducted by *Fomin, Gorokhov & Tymoshenko (2006)*, who found that the creatinine concentration in blood serum increased by 2–2.5 times in untrained men in response to physical stimuli (physical activity consisted of 40 intense squats) compared to data fixed in professional athletes (weightlifters, taekwonders). This fact indicates the low level of body resistance in untrained men, whose urgent adaptation occurs due to the creatine phosphokinase mechanism of energy supply.

The increase of creatinine concentration in blood serum of both group representatives in response to physical stimuli fixed after 3 months of training different in volume and intensity, confirms the results of studies conducted by several authors (*Shin et al., 2016*; *Stajer, Vranes & Ostojic, 2018*; *Omassoli et al., 2019*). A similar increase in creatinine concentration in response to physical stress stimuli is primarily associated with structural

damage of the working muscles and may subsequently lead to impaired renal function (e.g., *Chlíbková et al., 2019*).

Studying the nature of changes in the basal level of creatinine concentration we found out that this biochemical indicator increased in group B representatives due to accelerated growth of the body strength capabilities, circumference body size and, accordingly, muscle mass overall. The obtained data confirmed the results of experimental studies indicating that the greater is the muscle mass of a human body, the higher is the basal level of creatinine concentration in blood serum (*Fomin, Gorokhov & Tymoshenko, 2006*; *Butova & Masalov, 2011*; *Futorniy et al., 2016*).

Thus, the analysis of the results obtained during a series of experimental studies indicated the need and feasibility of using the index of basal level of creatinine concentration in blood serum during fitness training as an informative marker of assessing the process of long-term adaptation, especially in conditions of high intensity and low volume training load regime ($R_a = 0.72$).

## CONCLUSIONS

At the beginning of the study the creatinine concentration in blood serum decreased in response to physical stimuli compared with the state of rest, regardless of the training load regimes. These changes indicate a low basal level of body resistance to fitness training load in both groups and the inability to activate mechanisms of urgent adaptation to meet the increased need for energy supply using the creatine phosphokinase (alactate) resynthesis of ATP.

At the end of the 3rd month of research the index of the projectile working mass ($m$) was 47.9% higher in group B compared with group A; the indicator of amount of training load in a set ($Wn$) was 44.4% higher in the men of group A compared to group B. The creatinine concentration in blood serum (almost 80%) of the examined contingent showed a more pronounced increase in response to physical stimuli in conditions of long-term using low intensity training load regime ($R_a = 0.53$), indicating active usage of creatine phosphate in the process of energy metabolism.

Long-term usage of high intensity training load regime ($R_a = 0.72$) during fitness training increases the basal level of creatinine concentration in blood serum, thus indicating positive adaptation changes in the body and increasing the level of training and morphometric indices of the examined contingent due to muscle mass growth.

## ACKNOWLEDGEMENTS

The article is a part of the planned scientific work "Development and implementation of innovative technologies and correction of the functional state of a person during physical activity in sports and physical therapy", (state registration number 0117U007145) and Ministry of Education and Science of Ukraine project number 0118U000809.

### Funding
The authors received no funding for this work.

### Competing Interests
The authors declare there are no competing interests.

### Author Contributions
- Andrii Chernozub and Georgiy Korobeynikov conceived and designed the experiments, performed the experiments, analyzed the data, prepared figures and/or tables, authored or reviewed drafts of the paper, and approved the final draft.
- Vladimir Potop performed the experiments, analyzed the data, prepared figures and/or tables, authored or reviewed drafts of the paper, and approved the final draft.
- Olivia Carmen Timnea performed the experiments, analyzed the data, authored or reviewed drafts of the paper, and approved the final draft.
- Oleg Dubachinskiy performed the experiments, authored or reviewed drafts of the paper, and approved the final draft.
- Oksana Ikkert conceived and designed the experiments, performed the experiments, prepared figures and/or tables, reagents/materials/analysis tools, and approved the final draft.
- Yuriy Briskin and Yuriy Boretsky conceived and designed the experiments, performed the experiments, authored or reviewed drafts of the paper, and approved the final draft.
- Lesia Korobeynikova conceived and designed the experiments, performed the experiments, analyzed the data, prepared figures and/or tables, and approved the final draft.

### Human Ethics
The following information was supplied relating to ethical approvals (i.e., approving body and any reference numbers):

The experimental study was approved by Petro Mohyla Black Sea National University Ethics Committee for Biomedical Research in accordance with the Ethical Standards of the Helsinki Declaration (ecbr5-10-2019). The participants gave written consent to the study in accordance with the recommendations of the Biomedical Research Ethics Committees (*WHO, 2000*).

### Data Availability
The statistical data are available in Supplemental File.

### Supplemental Information
Supplemental information for this article can be found online at http://dx.doi.org/10.7717/peerj.9137#supplemental-information.

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
