# Peer review of "Creatinine is a biochemical marker for assessing how untrained people adapt to fitness training loads"

_PeerJ, doi:10.7717/peerj.9137_

## Round 0.1 · original submission · Major Revisions

The reviewers have indicated several concerns with your work, which you should address in a revised version of the text.

·

Basic reporting

Whereas the text is generally well written, there are some grammar mistakes throughout the manuscript which should be corrected Some examples are in L113, L128-129, L116-117, L150-151, L216, L240 and L284.

Abstract:
L41: What does the term ‘PM’ mean?

Your introduction is logical and well structured although it needs more detail. Specifically, a brief description of the sport/activity used to implement the exercise in the participants is needed. Why power fitness was selected as the specific tool to develop this study? Stating that it is mainly anaerobic is important but providing more details is needed.
By the way, the term ‘power fitness’ is used to mention a specific sport/physical activity but it is not. At least it is not known as it has been named in the current manuscript in the international research community. May the authors change this term to ‘resistance training’ throughout the text?
L75: I think that authors refer to ‘high-intensity’ sports rather than ‘high-speed’ sports.
L83-86: If there are results they should be indicated although they would be scarce.

Experimental design

The methods are not sufficiently described.
L 107: What does the term ‘PM’ mean?
L108: The method of ‘control testing’ is just explained through a reference (Shaner et al., 2014) which is not even in the references section. An explanation of this method is needed.
L112: an explanation of the method used is needed.
L113-115: it is not clear which method of anthropometry measurement was used. It should be specified and explained.
L116: the kinetic method mentioned should be explained and referenced.
L121: A reference is needed here.
L124: Why ‘Ra=0.72’ and ‘Ra=0.53? These terms should be explained. How this coefficient is assessed?
L127: ‘reducing almost twice’? I think that this expression is not correct.
L124: What is the evidence for the use of this strategy of training load combination? Can this be based on the findings of a previous study?
L124: If in this paragraph, authors are referring to a modification of the training regime with respect to the control group, it should be clarified.
Considering that main group participants used a different
L150: I’m not sure whether the training characteristics of the main group reflects a higher intensity and a lower volume than the training characteristics of the control group. In this sense, reducing the length of the rest periods implies a decrease of intensity and not the opposite. Training intensity is also related to the movement velocity and in this case this velocity is lower in the main group. Therefore, I think that authors may need to value whether it may need to be necessary to change this expression (‘high volume-low intensity vs low volume-high intensity’).
L151: I don’t understand the reference to ‘conventional indicator of power fitness’, specially when referring to a comparison of different types of training load combinations. May the author clarify this aspect?
Which exercises were used? May the authors indicate a description of the exercises used? How many sets per session did each group conduct?
L151-155: This paragraph somehow repeats what was stated in a previous paragraph (L124-129). They may be rephrased.
L160: I think that the term ‘using well-known methods’ may need to be removed.
L164-165: I think that this sentence should be removed because it repeats part of the previous text.

Validity of the findings

Results.

L189: a mention to the table where this explanation can be checked might be worth.
L191: I don’t understand the expression ‘the number of which depend on the…’. Please, clarify.
L190-194: This paragraph is difficult to understand. May the authors clarify it?
L205-211: This paragraph pertains rather to the discussion section.
L219-228: This paragraph pertains rather to the discussion section.

Discussion and conclusion.

The discussion is well structured and explained although I miss some references related to the resistance training methods which induces muscle hypertrophy and the relationships with the methods used in the current study. More specifically, I am concerned with the relationship of training intensity and volume made by authors and their findings. In this sense, a comparison with existing studies may be worth.

I think that some of the conclusions are not summarizing the main findings of this study and they are relying on other aspects which does not really reflect what has been found in the current study.
L309-312: This sentence is not clear. Please rephrase.

Additional comments

The present study deals with a very important topic regarding creatinine related adaptations to different types of training load combinations in non-trained population. Its findings are interesting and it is generally well written although it contains some issues that should be addressed.
1. The most important issue is the lack of description of some methods of analysis (i.e., the method used to assess the strength capability).
2. The next important issue is the lack of a wider description of the training methods used in order to understand better what participants did. Furthermore, the concept of low training volume-high intensity vs high volume-low intensity is not clear with respect to the training characteristics described and they need to be clarified.
3. The next important issue is that the term used by authors (‘power fitness’) to define the type of physical activity practised by participants is not common in the international research community and it may involve misunderstanding among readers of this manuscript.
4. The last but not less important issue is that I miss some references in the discussion related to the resistance training methods which induces muscle hypertrophy and the relationships with the methods used in the current study.

Reviewer 2 ·

Basic reporting

this manuscript requires basic science reporting.

1. It's very hard to follow the manuscript. each sentence is too long and sometimes the whole paragraph ends with a single sentence. I would suggest the authors write the manuscript simple basic scientific language so that everyone in the scientific community understands.

Experimental design

I think original primary research is very interesting and comes under the aims and scope of the journal.

The research question which is addressed in this manuscript is very interesting and useful information for the assessment of adaptive body changes during power fitness training.

The experimental design was good and authors have followed all the ethics while performing the experiments.

Validity of the findings

The original findings in the manuscript are novel and will have an impact on the scientific community and society.

Along with the creatine levels, authors also have measured the adaptive body changes maximal muscle strength growth and circumference body size to correlate their results.

Additional comments

The manuscript titles as "Creatinine is one of the biochemical markers for assessing the processes of untrained people adaptation to power fitness training loads" focused on the to study the changes in creatinine concentration in blood serum as a marker for the assessment of adaptive body changes during power fitness training. The authors have nicely planned and executed the experiments. I have a few concerns regarding some of the points made in the manuscript. In my opinion, the manuscript looks better if the authors address the following issue I have.

1. t's very hard to follow the manuscript. Each sentence is too long and sometimes the whole paragraph ends with a single sentence. I would suggest the authors write the manuscript simple basic scientific language so that everyone in the scientific community understands.

2. There is no figure legends for Figure 1 and 2. Also no legends for Tables 1,2. it's hard to follow them without the legends.

3. some of the numbers in table 2 are very hard to follow because in the manuscript text authors have discussed these numbers as percentage increment for every parameter but in the table, they reported as numbers. I would suggest include these numbers in the text as well in brackets or include percentage increment numbers in table 2.

4. Authors haven't mentioned anywhere what is Ra, how they calculate to be 0.72 or 0.53?

5. The authors should include the significance of the work in the Abstract conclusion and also at the end of the discussion.

I think after careful consideration and going through the manuscript, the manuscript is ready to accept in this journal only after fixing errors.

Reviewer 3 ·

Basic reporting

no comment

Experimental design

Some references which were not written by English were hard to check experimental methods for reviewers or readers.

Validity of the findings

The results including tables and figures were limited to supporting the conclusion.

Additional comments

In the current manuscript, objective data and strong evidence in tables and figures were inadequate to support the conclusion. I suggest that authors should provide significant results and appropriate statistical analysis to confirm that the blood creatinine raising was accompanied by muscle mass increasing or training load etc.

Annotated reviews are not available for download in order to protect the identity of reviewers who chose to remain anonymous.

---

## Round 0.2 · Minor Revisions

Still pending some modifications suggested by one of the reviewers.

·

Basic reporting

After asking authors to include an explanation regarding the concept of 'power fitness', authors refused to provide it arguing that in Ukraine it is well known and failing to realize that the current journal has an international audience.

Experimental design

No comment

Validity of the findings

No comment.

Additional comments

Whereas authors have corrected some of the issues reported previously, they refused to provide an explanation regarding the concept of 'power fitness' arguing that in Ukraine it is well known and failing to realize that the current journal has an international audience.

Minor comments:
L88-91: Again, if there are few results they should be indicated. If there are not any studies related, it should be indicated rather than ‘few’.

Reviewer 2 ·

Basic reporting

the new manuscript much better in terms of the language and references and structure of the manuscript.

Experimental design

Methods and results are well described.

Validity of the findings

The original findings in the manuscript are novel and will have an impact on the scientific community and society.

Along with the creatine levels, authors also have measured the adaptive body changes maximal muscle strength growth and circumference body size to correlate their results.

Additional comments

The manuscript titles as "Creatinine is one of the biochemical markers for assessing the processes of untrained people adaptation to power fitness training loads" focused on the to study the changes in creatinine concentration in blood serum as a marker for the assessment of adaptive body changes during power fitness training. After careful consideration of the author's response to reviewers' comments, now the manuscript looks much better in terms of the data representation and conclusion. I would recommend accepting the manuscript with the current state. I don't have any comments now.

---

## Round 0.3 · accepted · Accept

All the reviewers' concerns have been correctly addressed.

·

Basic reporting

No comment

Experimental design

No comment

Validity of the findings

No comment

Additional comments

I think that the current study has substantially improved its quality during the review process and it is interesting for both practitioners and research community.